# Endoscope Automation Framework with Hierarchical Control and Interactive Perception for Multi-Tool Tracking in Minimally Invasive Surgery

**DOI:** 10.3390/s23249865

**Published:** 2023-12-16

**Authors:** Khusniddin Fozilov, Jacinto Colan, Ana Davila, Kazunari Misawa, Jie Qiu, Yuichiro Hayashi, Kensaku Mori, Yasuhisa Hasegawa

**Affiliations:** 1Department of Micro-Nano Mechanical Science and Engineering, Nagoya University, Furo-cho, Chikusa-ku, Nagoya 464-8603, Aichi, Japan; 2Institutes of Innovation for Future Society, Nagoya University, Furo-cho, Chikusa-ku, Nagoya 464-8603, Aichi, Japanhasegawa@mein.nagoya-u.ac.jp (Y.H.); 3Aichi Cancer Center Hospital, Chikusa Ward, Nagoya 464-8681, Aichi, Japan; 4Graduate School of Informatics, Nagoya University, Chikusa Ward, Nagoya 464-8601, Aichi, Japan

**Keywords:** autonomous systems, Hierarchical Quadratic Programming, robotic surgery, segmentation and tracking, visual servoing

## Abstract

In the context of Minimally Invasive Surgery, surgeons mainly rely on visual feedback during medical operations. In common procedures such as tissue resection, the automation of endoscopic control is crucial yet challenging, particularly due to the interactive dynamics of multi-agent operations and the necessity for real-time adaptation. This paper introduces a novel framework that unites a Hierarchical Quadratic Programming controller with an advanced interactive perception module. This integration addresses the need for adaptive visual field control and robust tool tracking in the operating scene, ensuring that surgeons and assistants have optimal viewpoint throughout the surgical task. The proposed framework handles multiple objectives within predefined thresholds, ensuring efficient tracking even amidst changes in operating backgrounds, varying lighting conditions, and partial occlusions. Empirical validations in scenarios involving single, double, and quadruple tool tracking during tissue resection tasks have underscored the system’s robustness and adaptability. The positive feedback from user studies, coupled with the low cognitive and physical strain reported by surgeons and assistants, highlight the system’s potential for real-world application.

## 1. Introduction

Minimally Invasive Surgery (MIS) has received great attention in the medical field, offering advantages such as reduced trauma and quicker recovery times compared to traditional surgical methods. Primarily studied in laparoscopic, thoracic, and colon operations [1], MIS has been expanding its range of applications due to advancements in surgical tools and techniques. With the introduction of robotic assistance, MIS is transforming into Robot-Assisted Minimally Invasive Surgery (RAMIS), aiming to overcome various limitations and challenges.

### 1.1. Background

Central to the success of RAMIS is the endoscope, a long, thin, and remotely controlled device that provides surgeons with a live video feed from inside the patient’s body. Proper positioning of the endoscope and other surgical instruments is crucial, and even a small misalignment can lead to complications [2]. Currently, surgical assistants manually handle the endoscope, guided by the lead surgeon’s verbal commands. However, this approach has drawbacks, such as assistant fatigue and the potential for misunderstanding, which compromise the field of view (FOV) control [3]. Hence, the development and automation of robot-assisted endoscope control systems are considered crucial steps for improving both the efficacy and safety of MIS [4]. Recent works in RAMIS worked on replacing the assistant by designing interfaces the robotic systems that allow to control the endoscope direct hand-operation [5], foot pedal [6], eye-tracking [7] or head-mount displays [8,9], etc. Nevertheless, these control schemes require specialized training and might result in higher mental load on the surgeons [3]. Hence, intelligent visual assistance through adaptive field of view adjustment is one of the crucial challenges that needs to be addressed to further increase the levels of autonomy in RAMIS. Addressing this issue is essential for expanding the deployment of RAMIS across a wider variety of medical cases [10].

For robust and safe use of teleoperated and autonomous RAMIS, multiple challenges need to be addressed [11]. In this context, we focus on the specific challenges related to endoscope automation, an essential component of general RAMIS. The endoscope, inserted through a trocar, must adhere to remote-center-of-motion (RCM) constraints for patient safety; this adherence can lead to surgical view misorientation, complicating the surgeon’s hand–eye coordination [12,13]. Moreover, the 2-D monocular vision common in endoscopes limits depth perception and complicates vision-based controls. Perception issues, like tracking surgical tools and estimating the viewing area or Region-Of-Interest (ROI), also need resolution to provide visual assistance.

### 1.2. Related Works

#### 1.2.1. Instrument Segmentation and Tracking

State-of-the-art techniques often utilize color markers for tracking surgical instruments, especially given their reflective, cylindrical surfaces [14,15]. These markers, however, are susceptible to contamination from factors like bleeding, thereby leading to algorithmic failures in tracking. As an alternative, YOLO-based detectors are increasingly employed [16]. Some research has advocated for specialized detectors that focus on generating image masks or precise tool poses [8,17,18,19]. Despite promising results, these methods are contingent on data availability and require frequent re-training for new surgical tools.

#### 1.2.2. RCM Constraint

Safety remains paramount in MIS. RCM constraints fulfill this role and are categorized into two primary types: mechanical and programmable [20,21]. Programmable RCMs offer more versatility and a broader range of adaptability. Approaches vary from geometric methods to optimization-based ones [22,23]. Due to RCM constraints, the degrees of freedom for manipulators are often reduced. This coupling between the end effector and image sensor complicates matters, making direct homogeneous transformations for hand-eye calibrations unfeasible. Several works address this challenge [13,14,24,25], but mostly under strong assumptions about detection accuracy and usually in single-tool scenarios.

#### 1.2.3. FOV Control in MIS

Different controllers aim primarily at minimizing misorientation error [14,15,24]. Recent studies consider multi-objective approaches or speech-based control as additional variables [25,26]. Null-space projection is commonly used to handle multiple constraints but can fall into local minima, particularly when dealing with inequality constraints like joint states. Optimization-based approaches offer a way to include these inequalities, although they come with the drawback of subjective weight tuning and lack of strict prioritization.

Substantial progress has been made in the development and control of multi-DOF flexible endoscopes [27,28,29,30]. These advancements enhance the FOV and mitigate the limitations found in rigid sensors.

Despite advances, significant gaps exist in the area of viewpoint and ROI adjustments. Most methods focus on single-tool tracking or employ data-driven methodologies [15,24]. Some even integrate surgeon preferences explicitly or via additional interfaces [8,25,26]. However, the extension of these methods for multi-tool handling is still an open research area.

### 1.3. Contributions

This paper investigates the automation of endoscopes within the specific context of tissue resection, which is one of the common surgical operations. In a minimally invasive setting, the surgeon and assistant must collaborate effectively while managing multiple tools confined within trocars, which presents a challenge for tissue manipulation. The dynamism of multi-user interactions within surgical procedures poses a unique challenge—one that has not been thoroughly addressed in current robotic surgery research.

We introduce a novel framework designed to enhance visual assistance and user experience in surgical settings. Our system ensures that both surgeons and assistants have access to visual information during operations, effectively reducing the potential impact of partial occlusions typical in MIS environments.

Our key contributions are:**Integration of HQP for Autonomous FOV Control:** We introduce a hierarchical framework that exploits Hierarchical Quadratic Programming (HQP) to autonomously control the endoscope’s FOV. This approach manages the multiple objectives inherent in surgical scenarios, ensuring an unobstructed view of the surgical field amidst the dynamic nature of multi-instrument manipulation.**Development of an Interactive Segmentation Module:** Central to our framework is what we refer to as the "Interactive Segmentation and Tracking Module." This provides surgeons flexibility in real-time during surgeries, allowing the selection and robust tracking of various objects within the surgical field. This module is instrumental for adapting to the dynamic and unpredictable conditions presented by multi-user surgical procedure.**Implementation Availability:** We provide a simulation environment and the corresponding codebase. Our contributions include the HQP-based control framework and the interactive segmentation and tracking modules, specifically designed for integration with robotic control systems. These resources aim to facilitate the integration of our system for practical applications in the future.

The remainder of the paper is structured as follows: Section 2 provides an overview of the proposed framework. Section 3 details the perception module and ROI strategies. Section 4 elaborates on the HQP-based controller design. Section 5 presents experimental validation and user case studies. Finally, Section 6 offers concluding remarks and directions for future work.

## 2. Proposed System Overview

The system architecture for endoscope FOV control is depicted in Figure 1. A central aspect of this system is that surgeons can selectively choose tools and objects of interest in specific surgical scenarios through the Graphical User Interface (GUI). The system utilizes the SAM model [31], a state-of-the-art segmentation algorithm, to extract masks for the selected targets. The Xmem model [32], a neural network-based tracking model, tracks the selected image masks, providing the capability to ensure that the targets remain visible even during interactions with tissues and other tools.

The system dynamically refines the ROI based on this tracked data, determining the optimal FOV for the ongoing procedure. By combining domain knowledge with specific task specifications, it discerns when the endoscope should zoom in or out, ensuring both precision and clarity in the surgeon’s field of vision. This integration guarantees the most critical elements are prioritized, enhancing the efficiency and accuracy of surgical endoscopy.

The endoscope’s FOV adjustment inherently involves multiple objectives, i.e., maintaining the RCM, tracking tools, aligning with the surgeon’s natural line of sight and adjusting the zoom level. Ensuring efficient visual assistance requires meeting these multiple constraints, which presents the challenge of designing controllers capable of satisfying all these constraints as the surgical interaction unfolds. A key feature of the proposed system is the HQP controller, designed to guarantee the satisfaction of the hierarchy of objectives and constraints, facilitating smooth and precise movements during FOV adjustments by a robotic assistant handling the endoscope throughout the surgical intervention.

The communications between the modules rely on the ROS (Robot Operating System) [33] interface, which enables the synchronization of sensor readings and motor commands, ensuring that only the latest readings are utilized for control.

## 3. Perception Methods

This section outlines the perception methods applied in our work. We detail interactive segmentation and tracking, discuss the underlying assumptions, and define ROI selection and adjustment strategies to provide visual assistance.

### 3.1. Interactive Segmentation and Tracking

In the dynamic environment of surgical procedures, the need for precise detection extends beyond surgical tools to various anatomical features within the endoscopic image. Developed through the training on over 1 billion annotations, predominantly sourced from natural images, SAM can deliver high-quality object masks. The model’s interactive design, which allows for user input through points, boxes, and textual prompts, significantly enhances its utility and adaptability [31]. In surgical applications, SAM’s capabilities hold promise for providing real-time overlays that accurately delineate anatomical structures and surgical instruments [34]. This can enhance a surgeon’s spatial awareness and provide the perception capabilities necessary for a robotic system to interpret various surgical scenes.

We enhance SAM’s capabilities by integrating it with Xmem, a neural network modeled on human memory processes and optimized for object tracking in video sequences. Xmem’s design is adept at managing memory usage efficiently, making it well-suited for real-time applications in surgery, where rapid changes and occlusions are commonplace.

The combined use of SAM for initial segmentation and Xmem for subsequent tracking ensures a robust workflow. Xmem maintains continuity by assigning persistent IDs to each object, thus enabling the system to keep track of tools that may get temporarily obscured. This feature is indispensable for maintaining an uninterrupted visual feed for the surgeon, which is paramount in complex surgeries involving multiple instruments.

Surgeons interact with this system via a GUI, as shown in Figure 2B, where they can select the objects of interest within the endoscopic view prior to the start of the surgery. SAM processes these selections, and once the segmented output, illustrated in Figure 2C, is approved, Xmem takes over to ensure continuous tracking throughout the procedure. As the surgical scene evolves, surgeons can re-select objects or introduce new ones as needed, offering flexibility for various scenarios without requiring re-training or re-calibration. The GUI is intentionally designed to facilitate easy re-selection and modification of tools and objects during a task, enhancing versatility for different research setups and prototype testing.

The efficacy of our tracking and segmentation system depends on the quality of the initial mask generated by SAM. While this study concentrates on the simultaneous tracking of up to four tools, the system’s architecture, underpinned by SAM and Xmem, but it can handle additional objects as needed. Those interested in the detailed performance metrics and capabilities of each model can find comprehensive analyses in their respective original publications [32,34].

### 3.2. Determining the ROI

Accurate estimation of the ROI is crucial to ensure the surgeon remains focused on surgical tasks. The ROI is dynamically centered on the collective barycenter of the surgical tools and the surgeon’s perspective within the endoscopic frame. The barycenter of each tool tip is extracted through image moment analysis.

For a given tool *i*, the barycenter $g_{i}$ is computed as follows:(1)$$g_{i}=\left(\frac{M_{10}^{\left(i\right)}}{M_{00}^{\left(i\right)}},\frac{M_{01}^{\left(i\right)}}{M_{00}^{\left(i\right)}}\right)$$
where $M_{10}^{\left(i\right)}$ and $M_{01}^{\left(i\right)}$ are the spatial moments of order one, and $M_{00}^{\left(i\right)}$ is the zeroth-order moment, or area, of the tool’s contour.

The system’s center of mass $\left(x_{m},y_{m}\right)$, which includes all the tools and the surgeon’s line of sight, is determined by assigning weights to these barycenters based on their significance to the operation.

Assuming the coordinates $\left(x_{g1},y_{g1}\right),\left(x_{g2},y_{g2}\right),\ldots ,\left(x_{gn},y_{gn}\right)$ for barycenters g1,g2,…,gn, and their corresponding weights $m_{1},m_{2},\ldots ,m_{n}$, the center of mass is calculated as:(2)$$x_{m}=\frac{\sum_{i=1}^{n}m_{i}x_{gi}}{\sum_{i=1}^{n}m_{i}},\;y_{m}=\frac{\sum_{i=1}^{n}m_{i}y_{gi}}{\sum_{i=1}^{n}m_{i}}$$

Upon aligning the endoscope’s lens center with the center of mass, the new camera center coordinates $\left(x_{c},y_{c}\right)$ are offset by (u,v) from the image plane’s top left corner:(3)$$x_{c}=x_{m}+u,\;y_{c}=y_{m}+v$$

In the absence of a tool, its weight is omitted from the center of mass calculation, allowing the ROI to adjust dynamically.

The radius of the ROI $r_{ROI}$ is derived from the minimum enclosing circles for each tool contours, taking into account the weighted distribution of the tools:(4)$$r_{ROI}=\max_{i\in \{1,2,\ldots ,n\}}\left\{\sqrt{{\left(x_{gi}-x_{c}\right)}^{2}+{\left(y_{gi}-y_{c}\right)}^{2}}+r_{eci}^{i}\right\}$$
for i=1,2,…,n, where $r_{eci}^{i}$ is the radius of the minimum enclosing circle for tool *i*.

The assignment of mass to each tracked tool can be informed by domain knowledge or empirical measures, such as the size of each object in the image approximated by the enclosing circle’s area. Figure 3 illustrates the ROI in a scenario where equal mass is assigned to two tools, along with estimated barycenters and approximate size estimation. Alternative methods may involve assigning priority based on the velocity or directionality of tool movement to enhance situational awareness and optimize visual assistance.

## 4. Hierarchical Framework for Surgical Robot Motion Planning

Autonomous control of surgical robots demands precise management of tasks such as maintaining a RCM, tracking visual features, aligning with the surgeon’s viewpoint, and adjusting magnification—all while operating within kinematic constraints. To efficiently address these tasks in real-time, a HQP controller is employed. This controller prioritizes tasks based on their priority, ensuring that the most important constraints are adhered to first, as depicted in the task hierarchy (Figure 1).

By structuring tasks in a hierarchical manner, HQP ensures that high-priority tasks, such as RCM adherence, take precedence over others. This approach allows for a systematic resolution of Quadratic Programming (QP) problems, ensuring safety and efficacy without compromising on the primary objectives of the surgery.

Finally, the modular nature of HQP allows for scalability and adaptability. As surgical techniques and tools evolve, new objectives can be integrated into the hierarchy, making HQP a robust and future-proof choice for the dynamic requirements of robotic-assisted surgeries [35]. This ensures that the endoscope’s field of view is optimally controlled, adapting to the complex needs of modern surgical environments.

### 4.1. HQP Controller

When a robot arm has more joints than needed to perform a task, it can use the extra degrees of freedom to do other tasks without affecting the main one. This idea allows us to create a list of tasks with different priorities and execute them in a hierarchical way using the robot’s redundancy. For a group of k tasks, each ranked in priority from 1 to *p*, the construction of the *k*-th task can be expressed as an optimization problem:
(5)$$\begin{array}{cc} \min_{x}\; & \frac{1}{2}\left|\right|A_{k}x-b_{k}{\left|\right|}^{2}\\ s.t.\; & C_{1}x\leq d_{1},...,C_{k}x\leq d_{k}\\ \; & E_{1}x=f_{1},...,E_{k}x=f_{k} \end{array}$$

Here x represents the optimization variable in $R^{n}$. Matrices $A_{k}\in R^{m_{k}\times n}$, $C_{k}\in R^{m_{{eq}_{k}}\times n}$, and $E_{k}\in R^{m_{{ineq}_{k}}\times n}$ along with vectors $b_{k}\in R^{m_{k}}$, $d_{k}\in R^{m_{{eq}_{k}}}$, $f_{k}\in R^{m_{{ineq}_{k}}}$ are arbitrary, where $m_{k}$ is the dimension of the *k*-th task and $m_{{eq}_{k}}$, $m_{{ineq}_{k}}$ are the dimensions of equality and inequality constraints.

The proposed HQP controller is capable of managing multiple tasks and constraints simultaneously, respecting the hierarchy between priority levels. It can also deal with tasks that have the same priority, allowing the introduction of soft constraints by assigning weights to each task within the optimization problem. A general form of the multi-objective problem given a priority level *p* is formulated as:(6)$$\begin{array}{cc} \min_{x,w}\; & \sum_{i=1}^{\eta_{p}}\frac{K_{t_{i}}}{2}\left|\right|A_{i}x-K_{r_{i}}r_{i}{\left|\right|}_{2}^{2}+\frac{K_{d}}{2}{\left||x|\right|}_{2}^{2}+\frac{K_{w}}{2}{\left||w|\right|}_{2}^{2}\\ s.t.\; & \;\;\;C_{p}x-d_{p}\leq w \end{array}$$
where $r_{i}$ is the *i*-th task residual, and $K_{t_{i}}$ and $K_{r_{i}}$ denote positive weights for task *i*. The variable w incorporates the inequality constraints, and the squared norm of x is used for regularization. Additionally, $K_{d}$ and $K_{w}$ represents positive weights for the regularization and slack terms, respectively. $C_{p}$ and $d_{p}$ correspond to a general matrix and vector, that describe the inequality constraints for tasks of priority *p*. To handle regularization issues with respect to singularities, a closed-loop IK scheme is employed, by including a proportional residual as optimization error given by $K_{r_{i}}r_{i}$ [36].

For each priority level, the optimization problem in the above equation is represented and solved as a QP problem. The order of priority is preserved by applying optimality conditions between consecutive tasks, as suggested by Kanoun et al. [37]. Lower-priority tasks are prevented from affecting higher-priority tasks by using the null-space projector operator in the optimization problem, ensuring that higher-priority tasks are executed without interference. The optimization problem, taking into account these optimality conditions, is:(7)$$\begin{array}{cc} \min_{x,w}\; & \sum_{i=1}^{\eta_{p}}\frac{K_{t_{i}}}{2}\left|\right|A_{i}\left(N_{p-1}x+x_{p-1}^{*}\right)-K_{r_{i}}r_{i}{\left|\right|}_{2}^{2}+\frac{K_{d}}{2}{\left||x|\right|}_{2}^{2}\\ \; & +\frac{K_{w}}{2}{\left||w|\right|}_{2}^{2}\\ s.t.\; & \;C_{p}\left(N_{p-1}x+x_{p-1}^{*}\right)-d_{p}\leq w\\ \; & \;C_{p-1}\left(N_{p-1}x+x_{p-1}^{*}\right)-d_{p-1}\leq w_{p-1}^{*}\\ \; & \;\;\;\;\;\vdots \\ \; & \;C_{1}\left(N_{p-1}x+x_{p-1}^{*}\right)-d_{1}\leq w_{1}^{*} \end{array}$$
where $N_{p-1}$ corresponds to the null space projector of the higher priority level with $N_{0}=I$ and $x_{0}^{*}=O$.

Each optimization problem can be addressed as a Quadratic Programming (QP) problem, in which:(8)$$\begin{array}{cc} z_{p}^{*}=\min_{z}\; & \frac{1}{2}z^{T}Q_{p}z+c_{p}^{T}z\\ s.t.\; & \bar{C_{p}}z-\bar{d_{p}}\leq 0 \end{array}$$
where the optimization variable is given by z=[xw], $Q={\bar{A_{p}}}^{T}\bar{A_{p}}$ and $c=-{\bar{A_{p}}}^{T}\bar{b_{p}}$. The matrices $\bar{A_{p}}$, $\bar{C_{p}}$ and vectors $\bar{b_{p}}$, $\bar{d_{p}}$ are obtained from the task formulation (Equation (Equation 7)) as
(9)$$\bar{A_{p}}=\left[\begin{array}{cc} A_{p}N_{p-1} & 0\\ 0 & K_{w}^{1/2}I \end{array}\right]\;\;\bar{b_{p}}=\left[\begin{array}{c} b_{p}\\ 0 \end{array}\right]$$
(10)$$A_{p}=\left[\begin{array}{c} K_{t_{1}}^{1/2}A_{1}\\ \vdots \\ K_{t_{n}}^{1/2}A_{n}\\ K_{d}^{1/2}I \end{array}\right]\;\;\;b_{p}=\left[\begin{array}{c} K_{t_{1}}^{1/2}\left(A_{1}{\dot{q}}_{p-1}^{*}-b_{1}\right)\\ \vdots \\ K_{t_{n}}^{1/2}\left(A_{n}{\dot{q}}_{p-1}^{*}-b_{n}\right)\\ O \end{array}\right]$$

The optimal solution ${\hat{x}}_{p}^{*}$ is then obtained by
(11)$${\hat{x}}_{p}^{*}=N_{p-1}x_{p}^{*}+{\hat{x}}_{p-1}^{*}$$
where ${\hat{x}}_{p-1}^{*}$ is the optimal solution for the previous p−1 optimization problem with higher priority.

### 4.2. Tasks Hierarchy

Our framework consists of various objectives, each with joint and velocity constraints. These objectives include respecting RCM constraint, visual servoing, alignment, and zoom level. To incorporate these tasks into the proposed HQP controller, we write them as QP problems.

#### 4.2.1. Remote Center of Motion

Given the insertion port’s position, denoted as $p_{\mathrm{trocar}}\in R^{3}$, and the positions of the joints before and after the RCM location denoted as $p_{\mathrm{cam}}\in R^{3}$ and $p_{\mathrm{end}}\in R^{3}$, shown in Figure 4, the RCM point, designated as $p_{\mathrm{rcm}}$, is determined as the closest point on the endoscope axis to $p_{\mathrm{trocar}}$. Its position can be calculated as follows:(12)$$p_{\mathrm{rcm}}=p_{\mathrm{cam}}+\left(p_{r}^{T}{\hat{p}}_{s}\right){\hat{p}}_{s},$$
where ${\hat{p}}_{s}=\frac{p_{\mathrm{end}}-p_{\mathrm{cam}}}{\left|\right|p_{\mathrm{end}}-p_{\mathrm{cam}}\left|\right|}$ represents the direction of the endoscope axis and $p_{r}=p_{\mathrm{trocar}}-p_{\mathrm{cam}}$ represents the difference between the trocar’s position and the endoscope base. The RCM error, denoted as $e_{\mathrm{rcm}}$, is defined as $\left|\right|p_{\mathrm{trocar}}-p_{\mathrm{rcm}}\left|\right|$.

The tool insertion port constrains the motion of the endoscope to 4 degrees of freedom, involving pitch and yaw pivoting around the RCM point, as well as roll and translation along the endoscope axis. Assuming that the endoscope is initially inserted through the insertion port, an allowable twist ${}^{\mathrm{rcm}}\xi_{\mathrm{rcm}}\in R^{6}$ representing the linear and angular velocities of the RCM point expressed in the RCM frame {rcm}, is given by
(13)$${}^{\mathrm{rcm}}\xi_{\mathrm{rcm}}=\left[\begin{array}{c} 0\\ 0\\ {}^{\mathrm{rcm}}\nu_{{\mathrm{rcm}}_{z}}\\ {}^{\mathrm{rcm}}\omega_{{\mathrm{rcm}}_{x}}\\ {}^{\mathrm{rcm}}\omega_{{\mathrm{rcm}}_{y}}\\ {}^{\mathrm{rcm}}\omega_{{\mathrm{rcm}}_{z}} \end{array}\right]$$

The HQP optimization variable, designated as $x\in R^{4}$, is defined within the operational space, such that $x=\left[\begin{array}{c} {}^{\mathrm{rcm}}\nu_{{\mathrm{rcm}}_{z}}\\ {}^{\mathrm{rcm}}\omega_{{\mathrm{rcm}}_{x}}\\ {}^{\mathrm{rcm}}\omega_{{\mathrm{rcm}}_{y}}\\ {}^{\mathrm{rcm}}\omega_{{\mathrm{rcm}}_{z}} \end{array}\right]$.

#### 4.2.2. Visual Servoing

The relationship between image feature velocities and the endoscope motion is given by [38]
(14)$$\left[\begin{array}{c} \dot{s_{x}}\\ \dot{s_{y}} \end{array}\right]=\left[\begin{array}{cccccc} \frac{\lambda }{z} & 0 & \frac{-s_{x}}{z} & \frac{-s_{x}s_{y}}{z} & \frac{\lambda^{2}+{s_{x}}^{2}}{\lambda } & -s_{y}\\ 0 & \frac{\lambda }{z} & \frac{-s_{y}}{z} & \frac{-\lambda^{2}-{s_{y}}^{2}}{\lambda } & \frac{s_{x}s_{y}}{\lambda } & s_{x} \end{array}\right]{}^{\mathrm{end}}\xi_{\mathrm{end}}=J_{I}{}^{\mathrm{end}}\xi_{\mathrm{end}}$$
where ${}^{\mathrm{end}}\xi_{\mathrm{end}}=\left[\begin{array}{c} {}^{\mathrm{end}}\nu_{\mathrm{end}}\\ {}^{\mathrm{end}}\omega_{\mathrm{end}} \end{array}\right]\in R^{6}$

Given that frames {rcm} and {end} are rigidly connected, ${}^{\mathrm{end}}\xi_{\mathrm{end}}$ can be computed by
(15)$${}^{\mathrm{end}}\xi_{\mathrm{end}}={\mathrm{Ad}}_{{}^{\mathrm{end}}X_{\mathrm{rcm}}}{}^{\mathrm{rcm}}\xi_{\mathrm{rcm}},$$
where ${\mathrm{Ad}}_{{}^{\mathrm{end}}X_{\mathrm{rcm}}}=\left[\begin{array}{cc} {}^{\mathrm{rcm}}R_{\mathrm{rcm}} & {\left[{}^{\mathrm{end}}p_{\mathrm{rcm}}\right]}_{\times }{}^{\mathrm{end}}R_{\mathrm{rcm}}\\ O & {}^{\mathrm{rcm}}R_{\mathrm{rcm}} \end{array}\right]\in R^{6\times 6}$.

The optimization problem for centering the image to the field of view is defined as
(16)$$\begin{array}{cc} \min_{x,w}\; & \frac{1}{2}\left|\right|J_{I}{\mathrm{Ad}}_{{}^{\mathrm{end}}X_{\mathrm{rcm}}}H_{m}x-K_{s}\left(s_{des}-s_{act}\right){\left|\right|}_{2}^{2}+\frac{K_{w}}{2}{\left||w|\right|}_{2}^{2}\\ s.t.\; & \;x-\left[\begin{array}{c} \bar{\nu }\\ \bar{\omega } \end{array}\right]\leq \bar{w}\\ \; & \;\left[\begin{array}{c} \underline{\nu }\\ \underline{\omega } \end{array}\right]-x\leq \underline{w} \end{array}$$
where $H_{m}=\left[\begin{array}{c} O_{2\times 4}\\ I_{4\times 4} \end{array}\right]\in R^{6\times 4}$, $w=\left[\bar{w},\underline{w}\right]\in R^{8}$ and $\underline{x}=\left[\underline{\nu },\underline{\omega }\right]$ and $\bar{x}=\left[\bar{\nu },\bar{\omega }\right]$ denotes the lower and upper limits for the endoscope linear and angular velocities, respectively. The optimization problem can be formulated as Equation (Equation 6) with coefficients
(17)$$A_{s}=J_{I}{\mathrm{Ad}}_{{}^{\mathrm{end}}X_{\mathrm{rcm}}}H_{m}\;\;b_{s}=K_{s}\left(s_{des}-s_{act}\right)\;\;\;C_{s}=\left[\begin{array}{c} I_{4\times 4}\\ -I_{4\times 4} \end{array}\right]\;\;\;d_{s}=\left[\begin{array}{c} \bar{x}\\ -\underline{x} \end{array}\right]$$

The optimal value ${\hat{x}}_{s}$ ensures proper endoscope motion towards reducing the visual servoing error.

#### 4.2.3. Misalignment

In the context of MIS, it is crucial to minimize the misorientation effects introduced by the motion of the laparoscope to maintain an intuitive FOV for the surgeon. The concept of the Intuitive Virtual Plane (IVP) is employed as a virtual reference plane aligned with the surgeon’s natural line-of-sight in open surgeries [14]. The IVP is particularly useful because, in MIS, the center of the ROI does not necessarily correspond to a real physical plane where depth can be easily measured.

To compute the misalignment angle θ, which represents the deviation of the laparoscope’s view from the IVP, we use the rotation matrix of the endoscope’s current orientation, *R*, and compare it to a predefined reference plane, $R_{\mathrm{ref}}$. The reference plane $R_{\mathrm{ref}}$ is chosen before the surgery to match the IVP as closely as possible.

The rotation matrix *R* contains three orthogonal unit vectors $r_{x},r_{y},$ and $r_{z}$, where $r_{z}$ is the normal vector to the plane defined by the laparoscope’s camera view. The reference rotation matrix $R_{\mathrm{ref}}$ has corresponding vectors $u_{x},u_{y},$ and $u_{z}$, with $u_{z}$ being the normal vector to the IVP.

To determine θ, we project the vectors $r_{x}$ and $r_{y}$ onto the plane defined by $u_{z}$, resulting in vectors $p_{x}$ and $p_{y}$. The angle θ is then calculated by comparing these projected vectors to the reference vectors $u_{x}$ and $u_{y}$.

The misalignment angle θ can be calculated using the following equation:(18)$$\theta =\mathrm{sign}\left(n\cdot d\right)\cdot \arctan\left(\frac{∥n∥}{∥d∥}\right)$$
where:$n=r_{x}\times p_{x}+r_{y}\times p_{y}$ is the vector representing the sine component of the angle.$d=r_{x}\times p_{y}-r_{y}\times p_{x}$ is the vector representing the cosine component of the angle.sign(n·d) ensures that the angle θ is measured in the correct direction.

By continuously calculating and adjusting for θ, the system can dynamically align the endoscope’s FOV with the IVP, thus maintaining an intuitive visual perspective for the surgeon during MIS procedures.

Since the misalignment error is related to the roll angle, a misalignment task Jacobian can be defined as
(19)$$J_{\theta }=\left[0,0,0,0,0,1\right]$$
which can be represented as an optimization problem (see Equation (Equation 7)) with coefficients
(20)$$\;A_{\theta }=J_{\theta }H_{m}N_{s}\;\;b_{\theta }=K_{\theta }\left(\theta_{des}-\theta_{act}\right)-J_{\theta }x_{s}^{*}\;\;\;C_{\theta }=\left[\begin{array}{c} I_{4}\\ -I_{4} \end{array}\right]N_{s}\;\;\;d_{\theta }=\left[\begin{array}{c} \bar{x}\\ -\underline{x} \end{array}\right]-C_{\theta }x_{s}^{*}$$
where $N_{s}=I_{4}-A_{s}^{†}A_{s}\in R^{4\times 4}$ corresponds to the Null space projector of the visual servoing task. The solution obtained is then integrated to the optimal solution found for the visual servoing task as:(21)$${\hat{x}}_{\theta }=N_{s}x_{\theta }^{*}+{\hat{x}}_{s}$$
that verifies the optimality condition and ensures that the visual servoing task is not affected by the orientation adjustments.

#### 4.2.4. Depth

Control of the endoscope insertion depth allows for adjustments in the field of view area, providing zooming in and out options, and is considered as a subsequent task. A depth task Jacobian is defined as
(22)$$J_{d}=\left[0,0,1,0,0,0\right]$$
which can be integrated into an optimization problem (see Equation (Equation 7)) with coefficients
(23)$$A_{d}=J_{d}H_{m}N_{\theta }\;\;b_{d}=K_{d}\left(d_{des}-d_{act}\right)-J_{d}x_{\theta }^{*}\;\;\;C_{d}=\left[\begin{array}{c} I_{4}\\ -I_{4} \end{array}\right]N_{\theta }\;\;\;d_{d}=\left[\begin{array}{c} \bar{x}\\ -\underline{x} \end{array}\right]-C_{d}x_{\theta }^{*}$$
where $N_{\theta }=N_{s}\left(I_{4}-{\left(A_{\theta }N_{s}\right)}^{†}\left(A_{\theta }N_{s}\right)\right)\in R^{4\times 4}$ corresponds to the null space projector of the visual servoing task. The optimal solution is then computed as
(24)$${\hat{x}}_{d}=N_{\theta }x_{d}^{*}+{\hat{x}}_{\theta }$$
where $x_{d}^{*}$ corresponds to the solution found from the depth optimization problem.

The proposed HQP framework enables to meet all the defined constraints providing a suitable ${}^{\mathrm{rcm}}\xi_{\mathrm{rcm}}$, which is the integrated to compute the target pose ${}^{\mathrm{end}}X_{{\mathrm{end}}_{des}}$ as
(25)$${}^{\mathrm{end}}X_{{\mathrm{end}}_{des}}=\exp\left({\mathrm{Ad}}_{{}^{\mathrm{base}}X_{\mathrm{end}}}{\mathrm{Ad}}_{{}^{\mathrm{end}}X_{\mathrm{rcm}}}{}^{\mathrm{rcm}}\xi_{\mathrm{rcm}}\right){}^{\mathrm{base}}X_{{\mathrm{end}}_{act}}$$

### 4.3. Inverse Kinematics

In order to reach the desired target pose ${}^{\mathrm{end}}X_{{\mathrm{end}}_{des}}\in SE\left(3\right)$, the desired joint values are computed through an additional QP optimization problem with its current initial endoscope pose ${}^{\mathrm{base}}X_{{\mathrm{end}}_{act}}$ as initial guess.

The tracking task’s residual is computed as:(26)$$r_{{\mathrm{end}}_{\left(q\right)}}=\log\left(X_{{\mathrm{end}}_{des}}X_{{\mathrm{end}}_{act}}^{⊤}\right)$$
where the logarithm log:SE(3)→se(3) maps the pose from the Lie group SE(3) to twists in the se(3) [39].

The joint and velocity limits are defined as:(27)$$\begin{array}{c} q^{-}\leq q\leq q^{+}\\ -{\dot{q}}_{max}\leq \dot{q}\leq {\dot{q}}_{max} \end{array}$$
where $q\in R^{n}$ denotes the manipulator’s joint configuration, $q^{-}$ and $q^{+}$ represent the joint limits, and ${\dot{q}}_{max}$ denotes the maximum allowable joint velocities. By adding convenient slack variables $w=[w^{+}\;w^{-}]$, both constraints can be integrated into an optimization problem with inequality constraints given by
(28)$$\begin{array}{cc} \min_{\dot{q},w}\; & \frac{1}{2}{\left||w|\right|}^{2}\\ s.t.\; & \;\dot{q}-\bar{q}\leq w^{+}\\ \; & -\dot{q}+\underline{q}\leq w^{-} \end{array}$$
where $\underline{q}=\max\left(\delta t\left(q^{-}-q_{act}\right),-{\dot{q}}_{max}\right)$ and $\bar{q}=\min\left(\delta t\left(q^{+}-q_{act}\right),{\dot{q}}_{max}\right)$.

The Inverse Kinematics problem is then defined as a QP optimization problem:(29)$$\begin{array}{cc} \min_{\dot{q},w}\; & \frac{K_{t}}{2}\left|\right|J_{\mathrm{end}}\dot{q}-K_{r}r_{\mathrm{end}}{\left|\right|}_{2}^{2}+\frac{K_{d}}{2}\left|\right|\dot{q}{\left|\right|}_{2}^{2}+\frac{K_{w}}{2}{\left||w|\right|}_{2}^{2}\\ s.t.\; & \;\;\;C\dot{q}-d\leq w \end{array}$$
where $J_{\mathrm{end}}$ represents the configuration dependent Jacobian matrix of the {end} frame, $K_{t}$, $K_{r}$ and $K_{w}$ are positive constants, $C=\left[\begin{array}{c} I_{n}\\ -I_{n} \end{array}\right]$ and $d=\left[\begin{array}{c} \bar{q}\\ -\underline{q} \end{array}\right]$.

## 5. Experiments

This section evaluates our hierarchical framework through a series of tests. We first examine the system’s tracking performance with a single tool in controlled conditions, then extend the analysis to user case studies simulating tissue resection.

### 5.1. Experimental Setup

Our system employs a Kinova Gen-3 robotic arm with a 7-DOF, a 30-degree, 31 mm-long rigid endoscope equipped with a high-resolution (2048 × 1080) image sensor, and integrated lighting. We used standard surgical tools and a training phantom abdominal cavity, as shown in Figure 5A,B. The endoscope’s camera operates at 60 fps with intrinsic parameters estimated by camera calibration: $f_{u}=1167.6213$, $f_{v}=1176.005$, u=1008 and v=536.

The perception and control modules operated on a Linux Ubuntu 20.04 workstation with a robust Intel Core i9-10980E processor and 94 GB of RAM. For kinematic computations, transformations, and parsing the kinematic chain, we utilized the Pinocchio library (v. 2.6.10) [40]. The backend for the HQP solvers was supported by CASadi (v. 3.5.5) [41], while the OSQP (v. 0.5.0) [42] handled computations related to QP problems. These tools facilitated the efficient operation of our system, allowing for quick computations necessary for real-time control.

The detection algorithm ran on an Nvidia RTX 3090 GPU, leveraging pretrained models for both SAM and Xmem. The Xmem utilized default memory parameters, as outlined in [32], with an average detection time of 0.052 s for up to four tools.

All components communicated seamlessly through the ROS Noetic framework, ensuring tight integration between the detection module and the controller.

### 5.2. Single Tool Tracking

This experiment aimed to confirm the system’s precision under near-ideal perception conditions with a single tool. The setup involved a robotic arm with a manipulator navigating a pre-defined path inside an abdominal model, as depicted in Figure 6. Using an Optitrack motion capture system, we placed markers on the endoscope to track its position against the constraints defined.

The task involved moving a grasping tool along a path marked within the model. Starting from the center, the operator, via a GUI, directed the tool sequentially to points A, B, C, and D, as outlined in Figure 6C. The robot adjusted the endoscope’s insertion depth and FOV accordingly.

Throughout the tracking task, the system maintained the RCM constraint within a 5 mm margin. Initially, the tracking error was negligible and, though it varied with the movement of the tool as captured in Figure 7b, any deviations were promptly corrected. Notably, even when rapid tool movements induced errors up to 150 pixels, the system demonstrated resilience by consistently realigning to within the 50-pixel threshold. This indicates a well-tuned feedback loop where the system compensates for tracking errors dynamically, adapting to the tool’s speed and trajectory. The depth error, while showing the greatest fluctuation due to its lower priority in the hierarchy of tasks, also diminished to acceptable levels once the tool’s motion ceased. This controlled fluctuation and prioritization highlight the system’s capacity to handle tracking variability, a crucial feature when the target is in motion, thus ensuring that task hierarchy is maintained during transient shifts in the tool’s position.

The error dynamics visualized in Figure 7 confirm the hierarchical constraint prioritization, with RCM error never exceeding set limits and other errors managed sequentially to optimize the FOV.

### 5.3. User Studies

We conducted user studies involving two male medical professionals, aged 52 and 33, both experienced in robotic laparoscopic surgery, along with eight novice male participants aged between 24 and 33, engaging them in three distinct tasks designed to simulate surgical tissue interactions. These tasks were grouped into single-operator scenarios with dual-tool usage and dual-operator scenarios with quadruple-tool interaction, detailed across three cases (Figure 8).

#### 5.3.1. Task Scenarios


*Task 1: Single Operator with Dual-Tool Interaction*


In Task 1, a single operator was presented with a rhombus-shaped path composed of nodes *A*, *B*, *C*, and *D*, illustrated in Figure 8. The operator used grasping forceps and scissors, starting with the scissors positioned on a dashed line section. The dashed line was divided into equal sections to represent the allowable single cutting distance. The operator’s objective was to grasp a red point on the tissue at the end of each section before proceeding to cut along to the next. This task simulated the tensioning and precision required in resection. The robot’s role was to maintain the ROI within a 200–250 pixel radius in the FOV and and to limit positioning error to 50 pixels.


*Task 2: Dual Operator with Quadruple-Tool Interaction*


Task 2 increased in complexity with two operators representing a surgeon and an assistant, each handling two tools. Similar to Task 1, the operators engaged with a rhombus-shaped path, but this time the surgeon had to select from one of three available cutting lines, as depicted in Figure 8. The surgeon placed the scissors on the chosen dashed line section and grasped the red point to simulate tissue tensioning. The assistant was responsible for positioning their tools on subsequent red points, forming a right-angle triangle with the surgeon’s instruments, to replicate the tissue triangulation process. The robot was programmed to modify the FOV to ensure all four tools remained within a 300–350 pixel radius ROI, maintaining a positioning error below 50 pixels, as the surgeon and assistant collaboratively simulated a resection.


*Task 3: Tissue Resection with Medical Professionals*


The final task was exclusively designed and attempted by medical professionals due to its complexity, involving a phantom tissue resection procedure. The task involved dynamic tool positioning by both the surgeon and the assistant to mimic an actual surgical environment. The surgeon determined the resection path, while the assistant triangulated and tensed the tissue. Unlike the previous tasks, the points for grasping were not predefined, requiring the operators to freely move and communicate their tool positions to establish the desired tensioning and triangulation prior to cutting. The robotic assistant’s challenge was to deliver visual support by keeping a dynamic FOV with an ROI radius of 300 pixels and a positioning error under 50 pixels that adapted to the rapid and varied movements of the operators’ tools, ensuring visual information was available for precise tissue resection. The demonstration of this task can be found in the Appendix A.

In all tasks, the robot’s performance was measured by its ability to keep the ROI within a predefined pixel radius and the smoothness with which it could adjust the FOV in response to the operators’ movements. The performance data were quantified by recording the total manipulation time ($T_{total}$) and the endoscope following rate (FR), with the latter indicating how effectively the robot’s camera followed the operators’ movements within the desired visual field:(30)$$FR=1-\frac{T_{out}}{T_{total}}$$
where $T_{out}$ is the amount of the time when the tracking error is larger than the defined threshold of 50 pixels, that is ROI is not positioned within the desired region.

#### 5.3.2. Participants for User Studies

All participants assumed the role of User 1 (surgeon) to manipulate the surgical tools during the study. A single novice participant was consistently assigned the role of User 2 (assistant) across all sessions.

Prior to data collection, each participant engaged in a 30-min training session with the dual-tool setup, assisted by visual feedback from a stationary endoscope robot. This preparatory session aimed to familiarize participants with the manipulation of the surgical tools and the workflow of the tasks they would be performing.

Over the following three days, each User 1 performed a series of tasks—either with two tools (solo) or with four tools (in collaboration with User 2). Each task was repeated five times to ensure robust data collection. We specifically reserved the actual resection tasks for the medical professionals, given the high complexity, with the assistance provided by the novice acting as User 2.

The effectiveness of the participants’ interactions was evaluated by recording manipulation times and collecting subjective feedback through questionnaires.

The detailed experimental procedure is as follows:Each User 1 completes a 30-min training session with the tools on the day before the experiment.During the main study, User 1 performs the assigned task, interacting with a GUI to position the tools and initiate tracking.Upon selecting a path, User 1 manipulates the tissue with visual assistance from the endoscope robot until the task is completed. This is done five times, with the endoscope returning to its initial position after each iteration.Following the tasks, User 1 completes a survey designed to capture their subjective experience.For medical professionals only, additional insights are gathered through informal post-task interviews.

#### 5.3.3. Experiment Questionnaires

We designed two distinct questionnaires to evaluate the participants’ subjective experiences. The first questionnaire focused on the overall outcome, speed, confidence in visual assistance, adequacy of ROI, and smoothness of FOV adjustment—factors modeled after the methodology of Yang et al. [14].

The second questionnaire employed the NASA Task Load Index framework, developed by Hart and Staveland [43], to estimate the cognitive load and fatigue levels experienced by users during the tissue manipulation tasks within the proposed framework. This assessment provided insights into the user’s workload and the system’s design across varying scenarios.

#### 5.3.4. Results on Users’ Case Studies

In contrast to single-tool tracking, User 1 and User 2 were granted the freedom to maneuver the tools according to their judgment. This flexibility sometimes resulted in partial tool occlusions and temporary removal of the tool from the endoscope’s FOV, particularly when selecting appropriate insertion points on the abdominal cavity model. Consequently, the system was challenged with various noise factors: occlusions, changes in background and lighting conditions (adjusting the endoscope’s distance alters brightness, affecting perception quality), as well as alterations in the tools’ orientation, shape, and configuration during tissue interaction. Additionally, task completion speeds varied among participants. These variables combined to offer a robust set of scenarios for evaluating system performance.

Table 1 summarizes the experimental results across the three task scenarios. It shows variation in manipulation time for each case, highlighting the differing complexities and skill requirements of the tasks. It is important to note that the error values in the table represent the average of the mean error values for each participant’s performance.

The following rate (FR), expressed as a percentage, sheds light on the system’s performance in different scenarios. The FR averaged 81.4% for the two-tool scenario, 76.1% for the four-tool case, and 78.4% for the resection task. The smaller difference between the four-tool case and the resection task can be attributed to the medical professionals’ superior hand-eye coordination, resulting in more stable forceps motion.

The primary metric for visual servoing, the tracking error, varied between 90 and 113 pixels across all cases. The highest error was observed in the two-tool scenario, likely due to the ROI being significantly affected by hand motion and potential tool loss, making it more sensitive to manipulation quality and perception stability. The four-tool and resection task tracking errors were similar, averaging at 90 pixels, though the resection task showed higher variance due to the compact arrangement of tools and the lessened impact of individual forceps movement on the overall ROI position.

Additional errors related to constraints within the HQP controller were compared to single-tool tracking experiments. The RCM error remained under the specified threshold for all tasks, while depth error showed the most fluctuation, with a mean range of 0.007 m to 0.011 m across tasks.

The questionnaire outcomes, depicted in Figure 9 and Figure 10, reveal that the overall satisfaction across all three tasks is comparably high, averaging around 5.5 on a 7-point scale. However, the notable variance in the four-tool scenario indicates its relative difficulty. Speed metrics exhibit the largest variability; despite mean values approximately at 4.5, the large variance suggests that task speed may need fine-tuning according to the user’s skill level. This inference is supported by the low variance in smoothness scores for the resection task, which only medical professionals undertook.

Confidence and ROI metrics similarly averaged at 5.5, but the variability in confidence, especially in the two-tool task, implies that incorporating more task-specific domain knowledge could enhance the user experience.

Figure 10 presents the NASA-Task Load Index results, presenting the perceived difficulty and user experience of the tasks. The data indicates a moderate workload across the tasks, with the highest perceived load scoring at 45 out of 100. The two-tool task is rated lower on the scale, suggesting it is less mentally and physically demanding. In contrast, the resection task scores higher by demonstrating the extra effort required due to the task difficulty.

Despite the diverse task complexities, the consistency in workload scores indicates the system’s capability to offer visual assistance while keeping cognitive and physical loads low during tissue resection procedures. To enhance the user experience, particularly for the complex resection task, additional adjustments to speed settings and ROI selection criteria may be required. Such refinements will ensure that even in the most demanding scenarios, the workload remains within a moderate and manageable threshold.

## 6. Conclusions and Future Works

In this research, we have introduced a hierarchical control framework utilizing HQP in conjunction with interactive segmentation based on SAM and Xmem algorithms. Our framework stands out for its ability to autonomously adjust the FOV of an endoscope, ensuring visual clarity during the dynamic conditions of surgical procedures. The SAM component of our system offers interactive segmentation capabilities, allowing users to select and segment any object within the image via a user interface. This interaction is highly customizable and user-friendly. Meanwhile, the Xmem algorithm ensures consistent and accurate tracking of these segmented objects, even under challenging conditions related such as temporary obscurity or partial occlusion.

A distinctive feature of our approach is its operational readiness without the need for pre-training, which enables the inclusion and consistent tracking of any object, regardless of the varying light, background conditions, or changes in the object’s appearance due to interactions with tissue. Tested in complex scenarios, including single-user with dual-tool interaction and multi-user with four-tool coordination, our system has showcased its robustness by offering visual assistance. The proposed system adeptly managed various constraints, including RCM, alignment, and depth level, while preserving the endoscope’s field of view and granting full control to both the surgeon and assistant during tissue manipulation. This capability has been substantiated through empirical robot motion analysis and user feedback questionnaires, resulting in high user satisfaction and minimal mental and physical load when using the proposed system.

Future works will focus on broadening the ROI to include various objects within the surgical field, thus requiring the system to anticipate user actions and intentions with greater precision. Refining the HQP constraints to incorporate collision avoidance between the endoscope and surgical tools is another challenging issue for advancement. This addition will significantly enhance the safety and functionality of the framework. Additionally, our focus will be on incorporating domain-specific knowledge and machine learning to evaluate the user’s skill level. These adaptive capabilities will enable the system to offer personalized and intuitive visual assistance, thereby optimizing the surgical workflow to align with the expertise of the medical professionals.

## Figures and Tables

**Figure 1 sensors-23-09865-f001:**
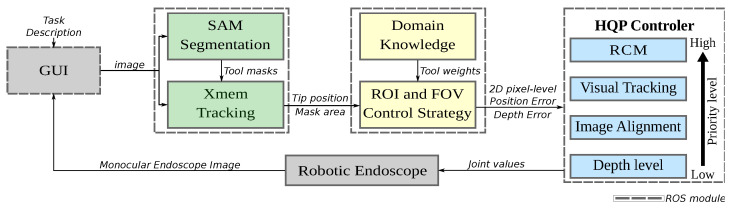
HQP based endoscope control system architecture.

**Figure 2 sensors-23-09865-f002:**
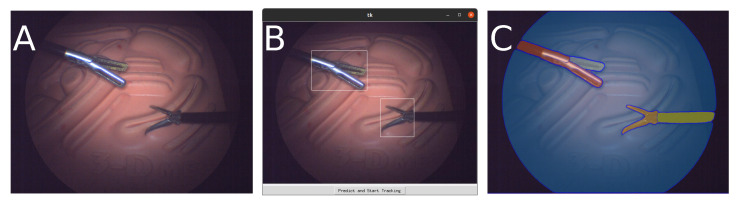
Illustration of interactive segmentation process: (**A**) Initial monocular endoscopic view, (**B**) User interface for choosing tools of interest, (**C**) Output displaying segmented regions post-SAM processing.

**Figure 3 sensors-23-09865-f003:**
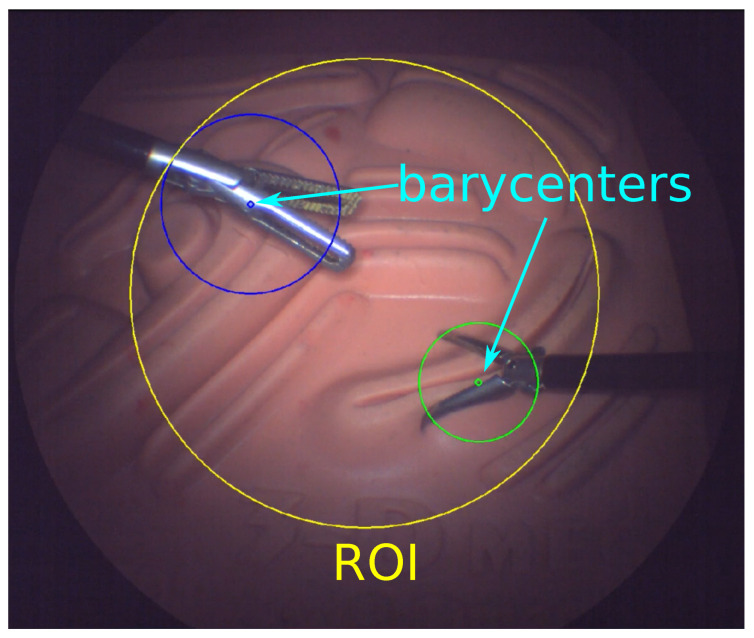
Estimation of tool barycenters and determination of the ROI center and radius based on the mass-weighted minimum enclosing circles. The extended blue and green lines visualize the $r_{eci}$ radii for the respective tools.

**Figure 4 sensors-23-09865-f004:**
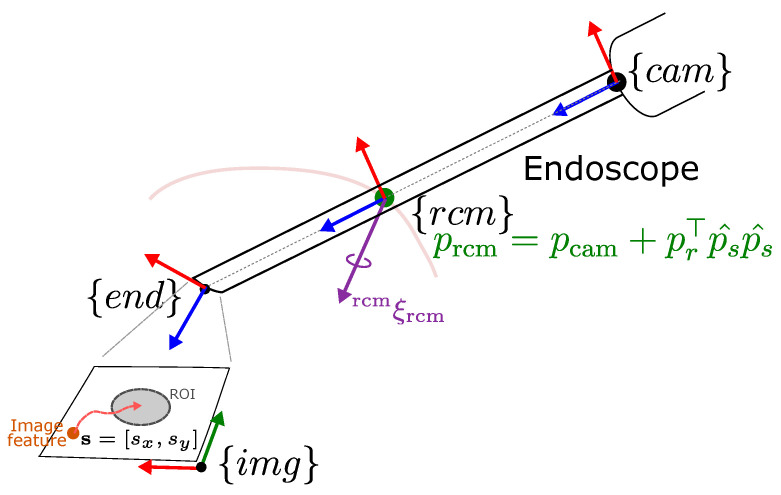
Coordinate frames for an endoscope motion control under RCM constraint.

**Figure 5 sensors-23-09865-f005:**
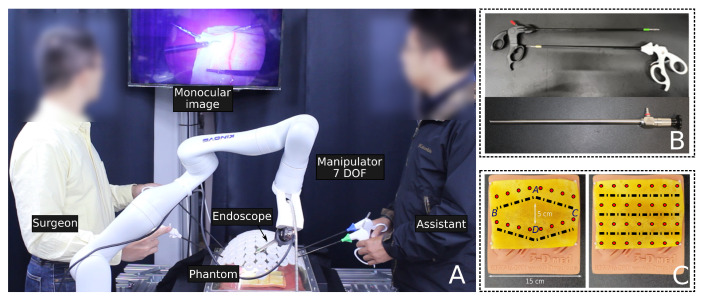
Experimental Setup Illustration: (**A**): The setup comprises a 7-DOF robotic manipulator holding a rigid endoscope in an environment designed for surgical procedures. Within this setting, a surgeon and an assistant collaborate to manipulate tissue inside a surgical phantom modeled after the human abdomen. They employ surgical instruments and rely on visual feedback displayed on a monitor to coordinate their actions. (**B**): Standard surgical training tools and a 30-degree endoscope used for experiments. (**C**): Tissue phantoms used for simulated tissue resection tasks.

**Figure 6 sensors-23-09865-f006:**
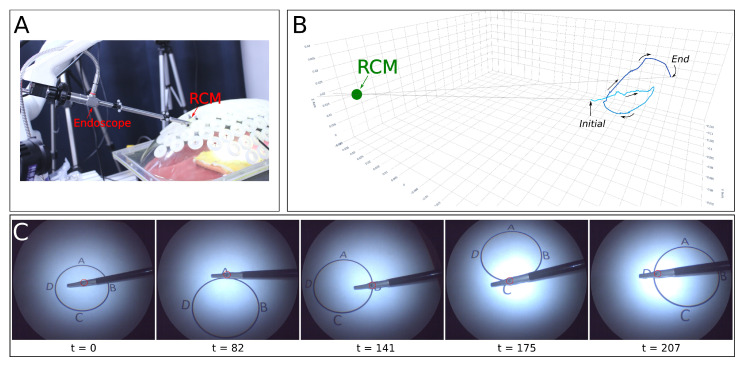
Experimental Setup for Single Tool Tracking: This illustration depicts the trajectory paths of the endoscope and the surgical tool, highlighting the coordination between visual tracking and instrument motion. (**A**): Overview of the setup, illustrating the attachment positions of markers for the Optitrack system. (**B**): The trajectory followed by the endoscope tip. (**C**): Sequential snapshots from the endoscope, revealing the tool’s movement during the experiment.

**Figure 7 sensors-23-09865-f007:**
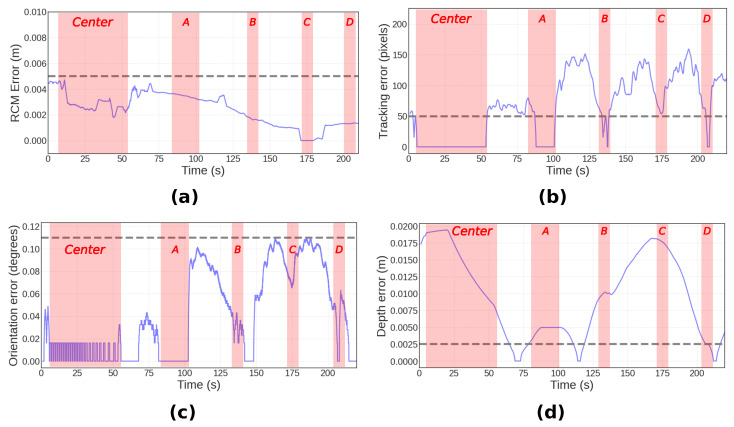
Variation in constraint errors during the single tool tracking experiment, illustrating the system’s hierarchy in managing constraints. (**a**): Change in RCM error (m). (**b**): Change in tracking error (pixels). (**c**): Change in misalignment error (deg). (**d**): Change in depth error (m).

**Figure 8 sensors-23-09865-f008:**
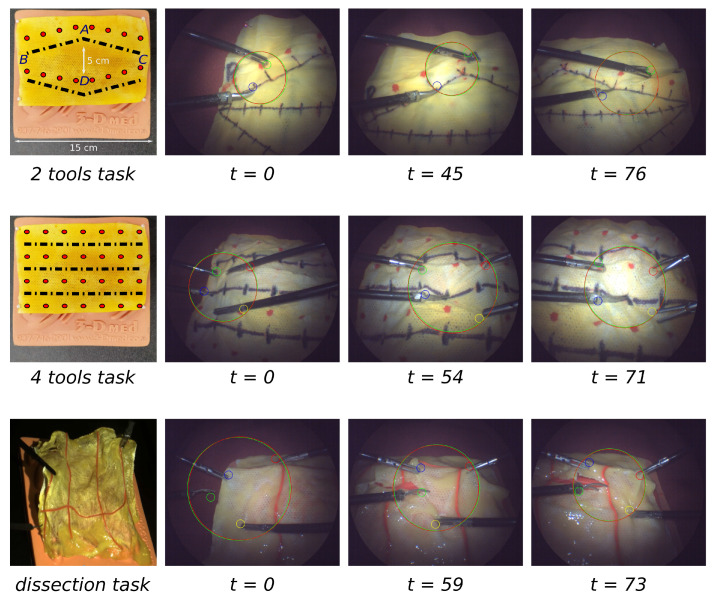
Snapshots from experiments (from the top), 2 tools, 4 tools and dissection tasks.

**Figure 9 sensors-23-09865-f009:**
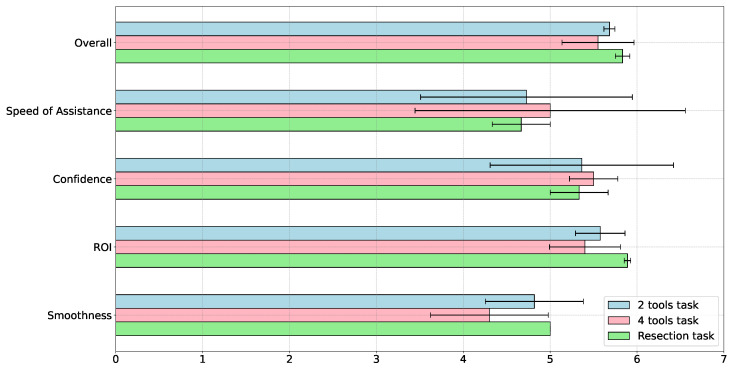
Comparative analysis of questionnaire results capturing user satisfaction and perceived task complexity for two-tool, four-tool, and resection tasks. Variance in scores highlights the differential difficulty levels and user experience across tasks.

**Figure 10 sensors-23-09865-f010:**
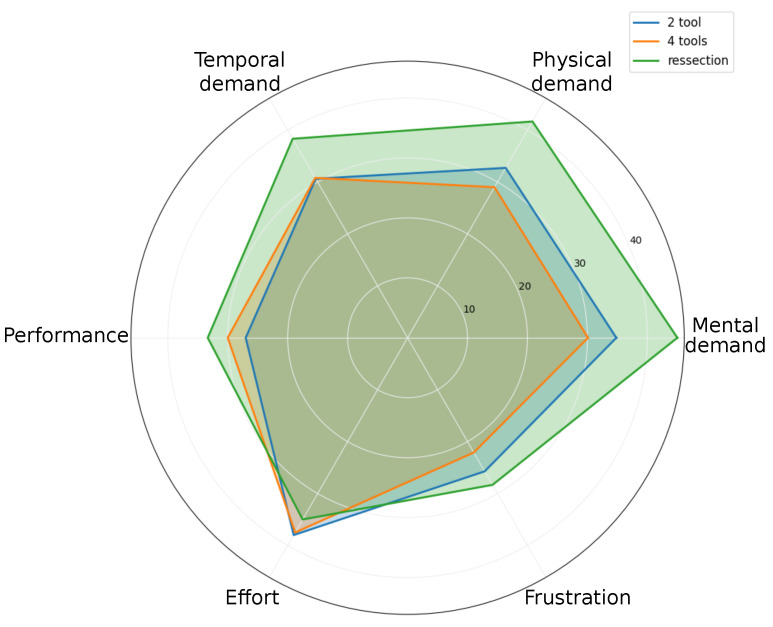
NASA-Task Load Index assessment profiles indicating cognitive and physical workload for the two-tool, four-tool, and resection tasks. Task load variations suggest differing levels of challenge and required user skill among the tasks.

**Table 1 sensors-23-09865-t001:** Performance metrics across two-tool, four-tool, and resection tasks, detailing manipulation time, following rate, and various error measurements. The data reflects the efficiency and precision of task execution, with standard deviations indicating variability among participants.

Metrics	2 Tools	4 Tools	Resection
Manipulation Time (s)	96.73 ± 10.34	164.27 ± 27.71	186.67 ± 21.59
Following Rate (%)	81.4 ± 21.7	76.1 ± 11.2	78.4 ± 22.9
Tracking Error (pixels)	113.30 ± 27.86	90.74 ± 38.48	91.68 ± 67.25
Angle Error (deg)	0.02522 ± 0.013196	0.01504 ± 0.01982	0.013599 ± 0.018781
Depth Error (m)	0.01108 ± 0.00439	0.00781 ± 0.00722	0.00974 ± 0.00799
RCM Error (m)	0.00279 ± 0.00181	0.00350 ± 0.00271	0.00431 ± 0.00311

## Data Availability

The implementation code of this work is openly available at https://github.com/husikl/asar_hqp_endoscope.git, accessed on 14 December 2023.

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
