# Peer review of "Endoscope Automation Framework with Hierarchical Control and Interactive Perception for Multi-Tool Tracking in Minimally Invasive Surgery"

_sensors, 2023, doi:10.3390/s23249865_

Round 1
Reviewer 1 Report
Comments and Suggestions for Authors
A framework is propsed that unites a Hierarchical Quadratic Programming (HQP) controller with an advanced interactive perception module. This integration addresses the need for adaptive visual field control and robust tool tracking in the operating scene. It ensures that surgeons and assistants have optimal viewpoint throughout the surgical task. User studies validate the framework’s adept handling of complex multi-tool MIS scenarios, demonstrating high user satisfaction and manageable workloads, highlighting its potential to improve surgical outcomes.
Suggest making modifications to the following sections. Firstly, it is recommended to modify and supplement the abstract and conclusion sections. Then, in the experimental section, it is recommended to add comparative testing and analysis with other methods.
Comments on the Quality of English Language Individual English descriptions need to be modified and improved.
Author Response
Summary of comment:
A framework is proposed that unites a Hierarchical Quadratic Programming (HQP) controller with an advanced interactive perception module. This integration addresses the need for adaptive visual field control and robust tool tracking in the operating scene. It ensures that surgeons and assistants have optimal viewpoint throughout the surgical task. User studies validate the framework’s adept handling of complex multi-tool MIS scenarios, demonstrating high user satisfaction and manageable workloads, highlighting its potential to improve surgical outcomes.
Thank you for your positive remarks. Below, we address your comments and queries.
Comment 1:
Suggest making modifications to the following sections. Firstly, it is recommended to modify and supplement the abstract and conclusion sections.
Response:
We appreciate your recommendation. The abstract has been updated to more accurately reflect the key results (page 1, lines 8-13):
“We have enhanced the abstract to detail how our framework efficiently manages multiple objectives within set thresholds, ensuring effective tracking amidst varying operating conditions such as changes in background, lighting, and instances of partial occlusions. The empirical validations, involving single, double, and quadruple tool tracking in tissue resection tasks, confirm the system's robustness and flexibility. Surgeons and assistants reported low cognitive and physical strain, underscoring the system's practical applicability and potential to enhance surgical outcomes.”
In the conclusion section (section 6, lines 550-557), we have incorporated additional details about the results, emphasizing the framework's unique contributions and advantages.
Comment 2:
In the experimental section, it is recommended to add comparative testing and analysis with other methods.
Response:
Thank you for this insightful suggestion. The related works section highlights that most existing studies focus separately on perception or control challenges. Our framework uniquely manages up to four tools simultaneously, offering more motion freedom to users, especially in complex interactions with tissues and under partial occlusions. This contrasts with existing methods like [25], which only facilitate sequential control of up to three tools by a single user. The distinct nature of our experimental scenarios, addressing these more complex and interactive challenges, makes direct comparisons with existing methods challenging. However, we believe this underscores the novelty and significance of our approach in advancing endoscope automation.
Reviewer 2 Report
Comments and Suggestions for Authors
The paper presents an innovative method for an autonomous endoscopic camera developed with the help of AI algorithms and Hierarchical Quadratic Programming controller. The paper's topic is very interesting and treats an actual challenge regarding the MIS procedures.
The paper is well-organized, presenting information in a clear and logical manner.
However, there are some aspects that I want to address to authors.
1. Please, insert in the text the references for the figures (2, 3, 5), in order to correlate the information with the visual representation of it.
Remain at this subject, take into consideration to rearrange the figure to be in vicinity of those paragraphs where the information is presented. As an example, the information in Figure 10 is located in a different section.
2. Consider providing additional details about the figures, such as Figure 1, which mentions the ROS framework. The integration of ROS into the system remains unclear to me. Although there is a brief explanation of using ROS on page 12, I find it insufficient and somewhat disconnected from the visual representation.
3. I recommend that the authors conduct additional testing on the system, exploring scenarios such as instances where the field of view is affected by smoke generated from electrocautery.
4. Please review the text to eliminate errors, such as redundant explanations for acronyms and repeated words (e.g., adoption and adoption). Ensure that acronyms have explanations only where required and avoid unnecessary duplications.
Author Response
Summary of comment:
The paper presents an innovative method for an autonomous endoscopic camera developed with the help of AI algorithms and Hierarchical Quadratic Programming controller. The paper's topic is very interesting and treats an actual challenge regarding the MIS procedures.
The paper is well-organized, presenting information in a clear and logical manner.
Thank you for your affirmative feedback and constructive suggestions.
Comment 1:
Please, insert in the text the references for the figures (2, 3, 5), in order to correlate the information with the visual representation of it.
Response:
We appreciate your guidance. References to figures 2, 3, and 5 have now been appropriately inserted at section 2 line 118, section 3 line 165, and section 5 line 362, respectively, ensuring better correlation between the textual content and visual representations.
Comment 2:
Remain at this subject, take into consideration to rearrange the figure to be in vicinity of those paragraphs where the information is presented. As an example, the information in Figure 10 is located in a different section.
Response:
Your observation is well-noted. We have realigned all figures, including Figure 10, to be in close proximity to the relevant text, enhancing readability and coherence between the figures and their descriptive sections.
Comment 3:
Consider providing additional details about the figures, such as Figure 1, which mentions the ROS framework. The integration of ROS into the system remains unclear to me. Although there is a brief explanation of using ROS on page 12, I find it insufficient and somewhat disconnected from the visual representation.
Response:
Thank you for this important feedback. We have updated Figure 1 to more clearly depict the integration of the ROS framework. The figure now features additional annotations and dashed lines to illustrate the communication pathways between individual ROS modules. Furthermore, we expanded the description in section 2 (lines 130-140) to include a more detailed explanation of ROS’s role in our system.
“The communications between the modules rely on the ROS (Robot Operating System) [33] interface, which enables the synchronization of sensor readings and motor commands, ensuring that only the latest readings are utilized for control.”
Comment 4:
I recommend that the authors conduct additional testing on the system, exploring scenarios such as instances where the field of view is affected by smoke generated from electrocautery.
Response:
We value your recommendation. While our initial tests focused on simplified scenarios with standard surgical training tools, we acknowledge the importance of testing in more complex and realistic conditions, such as those involving electrocautery smoke. Plans for such extended testing scenarios are outlined as part of our future directions.
Comment 5:
Please review the text to eliminate errors, such as redundant explanations for acronyms and repeated words (e.g., adoption and adoption). Ensure that acronyms have explanations only where required and avoid unnecessary duplications.
Response:
We are grateful for this attention to detail. The manuscript has been thoroughly reviewed and edited to remove any redundancies, repeated words, and erroneous acronym explanations. We have ensured that acronyms are explained clearly and only where necessary, improving the overall clarity and precision of the text.